# From IID to the Independent Mechanisms assumption in continual learning

**Oleksiy Ostapenko,**[1,2] **Pau Rodríguez,** [3] **Alexandre Lacoste** [3] **Laurent Charlin** [1,2,4]

[1] Mila - Quebec AI Institute, [2] Université de Montréal, [3] ServiceNow, [4] Canada CIFAR AI Chair

Current machine learning algorithms are successful in learning clearly defined tasks from large i.i.d. data. Continual learning (CL) requires learning without iid-ness and developing algorithms capable of knowledge retention and transfer, the later one can be boosted through systematic generalization. Dropping the i.i.d. assumption requires replacing it with another hypothesis. While there are several candidates, here we advocate that the independent mechanism assumption (IM) (Schölkopf et al. 2012) is a useful hypothesis for representing knowledge in a form, that makes it easy to adapt to new tasks in CL. Specifically, we review several types of distribution shifts that are common in CL and point out in which way a system that represents knowledge in form of causal modules may outperform monolithic counterparts in CL. Intuitively, the efficacy of IM solution emerges since: (i) causal modules learn mechanisms invariant across domains; (ii) if causal mechanisms must be updated, modularity can enable efficient and sparse updates.

**Setup.** We consider the observation space consisting of variables $X$ and $T$. We think of $T$ as a subset of observed input variables that carry information about the task to be performed (e.g. operations in a math equation), while $X$ caries contextual information (e.g. input digits) that can be thought of as an argument to the underlying causal mechanisms. Here we assume the setting of supervised learning, where the label $Y$ must be predicted from $X$ and $T$ – each observation is a tuple $(X, Y, T)$. Observations are sampled from the joint that factorizes as $p_t(Y, X, T) = p(Y|X,T)p_t(X,T) = \sum_Z p(Y|X,T,Z)p_t(X,T,Z)$, where $Z$ denotes a set of potentially unobserved attributes and $t$ is the time/task index. Such setting can be instantiated in the math equations domain similar to Mittal, Bengio, and Lajoie (2022): $X_1, X_2 \sim R^{[-1,1]}$, and $T$ describe the math operations to be performed (+/-/* etc.) (one or many operations per equation).

**The Independent mechanisms (IM) assumption** states that in causal factorization of the joint, the mechanism $p(Y|X,T,Z)$ contains no information about the causes $p_t(X,T,Z)$ and VV (Schölkopf et al. 2012). Hence, the true causal mechanism $p(Y|X,T,Z)$ is invariant across tasks

and environments. For simplicity, here we assume the independence of $X$, $Z$ and $T$: $p_t(X, T, Z) = p_t(X)p_t(T)p_t(Z)$.

The IM assumption can be extended to the mechanism $p(Y|X,T)$, which can be thought of as a composition of autonomous modules that operate independently from each other (Parascandolo et al. 2018; Goyal et al. 2019). That is, it can be approximated with a learnable function $f_\theta(\cdot)$ that is *compositional*. The most general definition of compositionality is that the meaning of the whole is a function of the meanings of its parts (Hirst 1992). We envision a model $f_\theta(\cdot)$ parameterized with a set of $M$ modules that compete with each other for explaining the current observation. The benefit of such system for CL is discussed next.

**Different distribution shifts.** Compositonal solutions can be useful under different types of distribution shifts in CL.

*Domain shift*: shift in the joint $p(X, Y, T)$ caused by shift in $p(X)$. Domain shift can be leveraged for learning causal mechanisms, that is the mechanism invariant across domains, under some structural assumptions (i.e. sparse change in the underlying graph). This principle is used by Arjovsky et al. (2019) for learning invariant (causal) representations. Perry, von Kügelgen, and Schölkopf (2022) showed that domain shifts can provide useful learning signal for identifying causal structures if the shift in the underlying causal graph is sparse. Importantly, domain annotation is needed for such learning, which is natural in CL—every detected distribution shift signals a new domain. Once the true mechanism is learned, faster generalisations to new domains is possible. Importantly, leveraging domain shift for learning causal mechanisms likely requires storing samples from seen domains in a replay buffer (Rolnick et al. 2019).

*New tasks*: shift in the joint $p(X, Y, T)$ caused by shift in $p(T)$ that introduces a new causal mechanism (e.g. math operation) to be learned by a new module. Existing CL methods like regularization (Kirkpatrick et al. 2017) of replay (Rolnick et al. 2019) applied to monolithic networks may perform on par with modular solutions under this shift in terms of forgetting. Later, however, should achieve better transfer and faster learning under the assumption that tasks share mechanisms (i.e. "+" is used in combination with two other distinct operation in two different task). Additionally, monolithic architectures have been shown to loose plasticity throughout CL (Dohare, Mahmood, and Sutton 2021) — a

drawback that may be mitigated through modularity.

***Hidden shift***: shift in the joint $p(X, Y, T)$ caused by shift in $p(Z)$. Consider an example, where the task is to interpret the meaning of a nodding gesture at some geographical location, that is unknown. When moving e.g from Canada to India the meaning of the nodding gesture can change while the meanings of other gestures (supposedly) may remain identical. In the example of math equations, a new environment can hypothetically change the meaning of the multiplication operation to, say, subtraction while not effecting other operations. Since $Z$ is unobserved, this drift requires sparse knowledge of a single mechanism without effecting other operations. Standard CL methods are likely to underperform in this setting, as old and new tasks become contradictory.

***Data amount shift***: knowledge about previously seen mechanisms needs to be updated as more training data becomes available. Modular architecture may be able to sparsely update only the affected modules, while a monolithic solution, with entangled mechanisms, would suffer from forgetting if no measures to prevent it are taken.

***Spurious correlation shift***: attributes correlate under $p_t$ but not under $p_k, t \neq k$. For example operation "+" has been seen together in the same equation with "-" in task $t$, which may result in routing mechanism of a modular solutions to mistakenly associate the high level variable "+" with the mechanism of subtraction. For modular solutions this shift might require updating only the routing mechanism, while monolithic one would require updating the whole net. The problematic of spurious features in the context of CL has been recently studied out by (Lesort 2022).

**How to learn modules representing true causal mechanisms** is hence an important open question. While several attempts have been made to design systems capable of discovering the true underlying data generative modules that comprise $p(Y|X, T)$ (Goyal et al. 2019, 2021; Parascandolo et al. 2018), there is no clear receipt to do it yet. Several inductive biases have been proposed, that facilitate learning of independent composable mechanisms, including competition (Parascandolo et al. 2018), information bottlenecks such as attention (Goyal et al. 2019) or functional bottlenecks (i.e. limiting the number of inputs a module can take) (Goyal et al. 2021; Ostapenko et al. 2022), or restricting modular communication to discrete variables (Liu et al. 2021).

**Preliminary result with Mixture of Experts (MoE).** Here, we design a simple attention based MoE model and train it continually on two streams of 5 and 7 tasks. In both streams $X$ is sampled uniformly from $R^{[-1,1]}$, and the corresponding $T$ (here task description is represented by a single variable) is samples uniformly from a set of predefined math operations. Labels $Y$ are generated by applying sampled mechanisms $T$ to the inputs $X$. Stream 1 represents new task shift (i.e. new operations are introduced with operations overlapping across tasks) and consists of 5 tasks ($t = 0...5$). First 5 tasks of Stream 2 are identical to stream 1, tasks 5 to 7 simulate the *hidden shift*. For example, the operation encoded in the input of $t = 5$ is addition, which is identical to $t = 0$ and $t = 1$, but the meaning of addition has changed from $x_1 + x_2$ to $(x_1 + x_2)/5$, which is

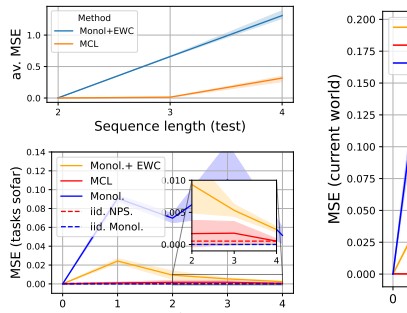

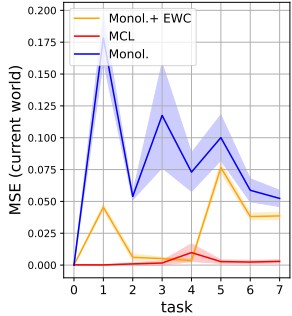

Figure 1: Stream 1.          Figure 2: Stream 2.

reflected in the training data of these tasks. The modular continual learner (MCL) receives as input a set of 3 entities: $x_1$, $x_2$ and an operation (e.g. addition, subtraction, multiplication etc.). All three variables are first projected into a vector space, thereby we use a fixed embedding table for the operations and an encoder, that is only trained during the first task, for $x_1$ and $x_2$. MCL performs module selection using key-value attention mechanism and a functional bottleneck similar to NPS (Goyal et al. 2021) (MCL is an adopted version of NPS for CL). We formulate these tasks as regression problems. We test on novel randomly sampled $x$'s. We use 20,000 samples per task for training and 2,000 for testing.

In Figure 1 we plot the average mean squared error (MSE) of all tasks after learning each task incrementally. MCL has a much larger mean MSE at the beginning, it reaches MSE comparable to EWC (Kirkpatrick et al. 2017) at the end of the sequence. In Figure 2 we measure MSE averaged over the current state of the world on Stream 2, i.e. if the meaning of "+" changes in $t = 5$ from $x_1 + x_2$ to $(x_1 + x_2)/5$, this change is incorporated in the test sets after this task (i.e. the addition operation in test sets of all tasks is replaced with addition and division by 5). Here, we observe that only MCL is able to perform well on this stream. EWC performs well up until $T_4$ as it is able to alleviate forgetting. After $T_4$, when the mechanisms shifts, EWC's regularization strategy, aimed at reducing plasticity, prevents the model from incorporating knowledge about the shift in the mechanism reflected in the new training data of tasks 5 to 7. MCL is able to sparsely update only the modules which are specialized on the shifted mechanisms. Importantly, MCL can eleviate forgetting solely through routing samples to correct modules.

**Conclusion.** We advocate for the usefulness of IM hypothesis in CL (it is not mutually exclusive with i.i.d). This may open a door for developing algorithms with better transfer and efficiency. We point out the potential advantages of such solutions under different distribution shifts and show in simple toy experiments that the IM principle can address some problems of CL in practice. Open challenges include determining useful inductive biases and further assumptions for designing modular solutions beyond MoE, where causal mechanisms can be discover when modules are applied in superposition resulting in a more fine-grained task decomposition (Ostapenko et al. 2022).

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
