# OpenReview forum: "From IID to the Independent Mechanisms assumption in continual learning"
_AAAI.org/2023/Bridge/CCBridge — AAAI23 Bridge Continual Causality_

### Official Review · Reviewer_gFxw · 2022-12-01
**Independent Mechanisms assumption for distribution shifts in continual learning**

**Rating:** 6
**Confidence:** 4

**Review:**

The authors proposed a formalism to understand the commonly encountered distribution shifts in continual learning. These shifts include new domains, new tasks, spurious correlations etc.

The mutual independence assumption of X, T and Z is rather stringent though. Coming for a pragmatic continual learning background, I'd suggest the authors to make more realistic assumptions.
In terms of results, MCL reaching the performance of EWC is not a very promising result.
The authors should also provide a bit more details in the figure captions.

---

### Official Review · Reviewer_AwxC · 2022-12-02
**Good paper with preliminary experiments**

**Rating:** 7
**Confidence:** 3

**Review:**

The paper studies the use of mixture of experts approaches for continual learning through the lenses of causal models. The Independent Mechanism assumption justifies the use of separate modules for different tasks. The preliminary results show promising performance on a simple task with arithmetic operations.
The link between the Independent Mechanism assumption in causality and the continual learning model is not completely clear. The model is partitioned into different experts, but the experts do not compute any causal relationship. It is not clear how can such a model learn causal representation and how that would contribute to the performance. If causality is only used to justify a Mixture of Experts approach, then the connection is rather weak. Nonetheless, the preliminary experiments remain valuable and may be worth exploring due to the final performance.

Minor note: What is VV at the end of the first column?

---

### Official Review · Reviewer_Z6Yi · 2022-12-02
**Outside of IID assumption in continual learning may is more interesting than plain IID.**

**Rating:** 9
**Confidence:** 3

**Review:**

The paper presents the continual learning with the distribution assumption of non-iid cases where they replay with independent mechanisms assumptions that integrate the causal mechanisms. It also tackles the replacement of the assumption impacts each factor in terms of continual learning with the brief background and toy practice. Overall, this paper not only fits the workshop goal but also opens up a new research domain for the community.

---

### Official Review · Reviewer_HAn3 · 2022-12-03
**Totally agree that the Independent Mechanisms assumption is a useful hypothesis**

**Rating:** 7
**Confidence:** 3

**Review:**

I totally agree that the Independent Mechanisms assumption is a useful hypothesis for representing knowledge in continual learning.

Pros:
1.  Independent Mechanisms assumption should be a promising direction.
2. Proposed a good abstract setup for various continual learning tasks, as well as some interesting discussion on different distribution shifts.
3. Provided some preliminary results in this direction.

Cons:

1. I am a little confused about the meaning of T in different  shifts.
2. One suggestion is that the author should consider the causal structures during the discussion in different shifts.

---

### Decision · Program_Chairs · 2022-12-05

**Decision:**

Accept

**Comment:**

Accept - Oral

The paper advocates for the use of modularity to adapt in a continual learning setting, motivated by the independent mechanism assumption. The paper is highly relevant to the bridge program and provides promising initial results. While the independent mechanism assumption is loosely motivated from a casual perspective, we encourage the authors to include a more detailed discussion of the causal aspects of their framework. We also encourage the authors to clarify the meaning of the variables T and W.